# Tailoring spin defects in diamond by lattice charging

Felipe Fávaro de Oliveira[1], Denis Antonov[1], Ya Wang[1], Philipp Neumann[1], Seyed Ali Momenzadeh[1], Timo Häußermann[1], Alberto Pasquarelli[2], Andrej Denisenko[1] & Jörg Wrachtrup[1,3]

Atomic-size spin defects in solids are unique quantum systems. Most applications require nanometre positioning accuracy, which is typically achieved by low-energy ion implantation. A drawback of this technique is the significant residual lattice damage, which degrades the performance of spins in quantum applications. Here we show that the charge state of implantation-induced defects drastically influences the formation of lattice defects during thermal annealing. Charging of vacancies at, for example, nitrogen implantation sites suppresses the formation of vacancy complexes, resulting in tenfold-improved spin coherence times and twofold-improved formation yield of nitrogen-vacancy centres in diamond. This is achieved by confining implantation defects into the space-charge layer of free carriers generated by a boron-doped diamond structure. By combining these results with numerical calculations, we arrive at a quantitative understanding of the formation and dynamics of the implanted spin defects. These results could improve engineering of quantum devices using solid-state systems.

[1] 3rd Institute of Physics, Research Center SCoPE and IQST, University of Stuttgart, Stuttgart 70569, Germany. [2] Institute of Electron Devices and Circuits, University of Ulm, Ulm 89081, Germany. [3] Max Planck Institute for Solid State Research, Stuttgart 70569, Germany. Correspondence and requests for materials should be addressed to F.F.d.O. (email: f.favaro@physik.uni-stuttgart.de) or to A.D. (email: a.denisenko@physik.uni-stuttgart.de).

Spin impurities in solids rank among the most prominent quantum systems to date[1–3]. Their established quantum control makes them leading contenders in quantum communication, computation and sensing. Particularly, spin defects hosted in diamond, similar to the nitrogen-vacancy (NV) centre, have proved to be versatile atomic-sized spin systems, with remarkable applications in quantum optics[4], information processing[5–7] and quantum sensing[8–11]. The negatively charged NV centre has an electron spin ($S = 1$) that shows a bright and stable spin-state-dependent photon emission under constant off-resonance illumination (that is, 532 nm wavelength), allowing spin-selective optically detected magnetic resonance at single sites under ambient conditions[12]. Recently, NV centres were successfully integrated into photonic[13–15] as well as mechanical[16,17] structures, further highlighting its versatility. Furthermore, NV centres hosted in isotopically purified ($^{12}$C) materials have demonstrated extremely long coherence times under ambient conditions ($\sim$ ms)[18], raising the NV centre in diamond as a potential solid-state spin system to fulfil the DiVinzenzo criteria for quantum computation and related applications, such as quantum sensing[19].

For most applications, NV centres must be created with nanometre spatial accuracy while still retaining excellent spin and optical properties. This is usually achieved by implanting nitrogen atoms with low energies ($<10$ keV) followed by thermal annealing. This technique accomplished the creation of NV centres with spatial resolution in the range of 5–10 nm for $<10$ keV implantation energy[20,21]. Despite its excellent positioning accuracy, the main disadvantage of low implantation energy is the concomitant low efficiency in the conversion from implanted nitrogen atoms to NV centres (yield) while the resulting NV centres suffer from degraded spin and optical properties[21,22]. This has been attributed mainly to residual implantation-induced damages in the host lattice formed in close vicinity to the implanted NV centres that are not fully eliminated by thermal annealing[23]. As previously pointed out, the degraded properties of the electron spin of implantation-induced NV centres limit their interaction time with target spin systems[9,24], which is the key roadblock to NV-based quantum applications. Understanding the dynamics and the formation of spin defects hosted in diamond is therefore of extreme importance, having also a broader implication for other solid-state defect-host materials, such as SiC[25] and rare-earth doped crystals[26].

Here we describe a method to tailor the formation dynamics of implantation-induced defects, accomplished by implanting nitrogen ions into a space charge layer of free carriers (holes) in an undoped substrate generated by a nanometre-thin boron-doped diamond layer on the sample surface. We show that the induced excess of free charge carriers in the diamond substrate changes the charge state of implantation defects with concomitant changes in their diffusion and recombination behaviour. Under such conditions, the formation of thermally stable defects with paramagnetic spin properties such as di-vacancy ($V_2$) complexes is strongly suppressed, as confirmed by a detailed analysis of the spin noise of near-surface NV centres. The obtained results are supported by numerical simulations of the formation and evolution of implantation-induced defects at single implanted ion sites. We demonstrate tenfold-improved spin coherence times ($T_2$) and spin-lattice relaxation ($T_1$) times $>5$ ms for single NV centres with depths of 2–8 nm below the diamond surface. These values are limited mostly by magnetic noise from surface spins. Furthermore, a twofold-improved yield is observed,

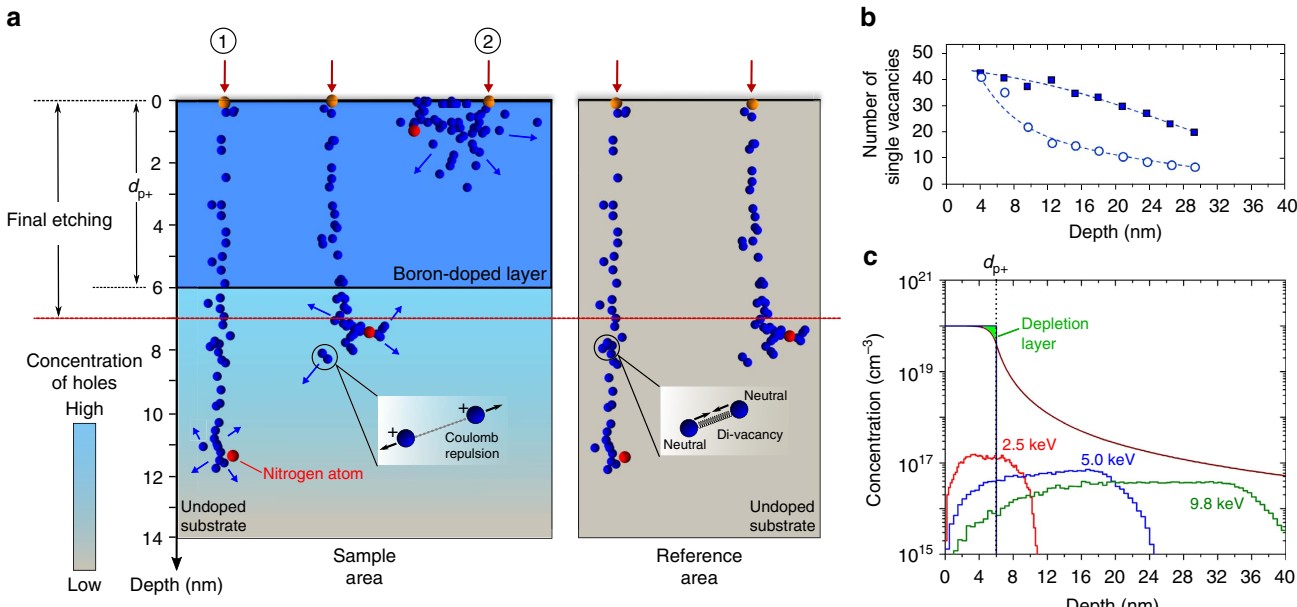

**Figure 1 | Simulated implantation defects in diamond and the proposed p$^+$–i structure.** (**a**) Individual implantation sites (indicated by red arrows) simulated by MD with 4.0 keV nitrogen kinetic energy are shown representing different contributions of ion channelling. Blue dots represent implantation-induced vacancies, whereas red dots represent the rest position of implanted nitrogen atoms. A schematic representation of the novel method of nitrogen implantation across a planar p$^+$–i junction is shown with the corresponding experimental steps (see left): boron-doped surface layer (6 nm) produced by CVD overgrowth, nitrogen implantation followed by thermal annealing and the final plasma etching step to remove the sacrificial boron-doped layer (7 nm)—the final fabricated region is named as 'Sample area' in the main text. The concentration of free charge carriers (holes) in the substrate is represented by the blue–grey colour scale. The reference area is also shown, where vacancies are found mainly in the neutral charge state. (**b**) Depth dependency of single vacancies counted at each implantation site (squares) or confined in defect clusters of nanometre radius (see Supplementary Note 1) around nitrogen atoms at rest (circles) extracted from the MD simulations. (**c**) 1D depth profile of boron acceptors (blue-solid line) and free holes (dark red-solid line) across the p$^+$–i junction structure at 950 °C by numerical simulation. Profiles of implanted nitrogen atoms for three different energies (2.5, 5.0 and 9.8 keV) simulated by CTRIM are shown for 3°-off implantation angle in a [100]-oriented diamond lattice (ion fluences of $10^{11}$ cm$^{-2}$).

indicating that the formation of NV centres occurs preferentially when the implanted nitrogen atoms occupy interstitial lattice positions, in agreement with recent results on the kinetics of NV centres upon thermal annealing[27].

## Results

**Formation dynamics of NV centres.** The kinetics of low energy ($<10$ keV) ion implantation is dominated by energy loss through nuclear interactions between the implanted ion and the host lattice[28]. As a consequence, implantation-induced vacancies are distributed along the entire trajectory of the implanted ion with small deviations due to different contributions of ion channelling. This is in contrast to what has been observed for higher implantation energy ($\sim$ MeV), where electronic energy loss dominates[29]. We start by investigating the distribution of vacancies at individual implantation sites by Molecular Dynamics (MD) simulations. Individual nitrogen implantation events are simulated for a kinetic energy of 4.0 keV in a [100]-oriented diamond lattice (see Supplementary Note 1 for further details). Implantation trajectories of nitrogen ions in diamond with identical implantation energies, but different ion implantation depths, are shown in Fig. 1a. Implantation sites labelled as (1) and (2) represent cases of pronounced presence and absence of ion channelling, respectively. As seen in the two cases, the spatial distribution of the as-implanted vacancies is strongly influenced by ion channelling.

The number of vacancies created per implanted ion shows similar values when integrated over the whole implantation path, as shown in Fig. 1b (squares). Nonetheless, the size of the resulting defect-cluster in a nanometre region surrounding the nitrogen atom at the end of the trajectory reduces with the implantation depth. This results in smaller numbers of vacancies in the immediate vicinity around the nitrogen atom at larger depths. Figure 1b shows a quantitative comparison of the number of vacancies confined around the nitrogen atom (circles) as a function of implantation depth. Low implantation energies lead to typical distances between single vacancies $<1$ nm, corresponding to local concentrations of $\sim 10^{21}$ cm$^{-3}$. In contrast, ultra-pure diamond substrates with $\sim 10^{14}$ cm$^{-3}$ concentration of nitrogen impurities[23,30] result in concentrations of free electrons of approximately $10^{13}$ cm$^{-3}$ at high temperatures. Under these conditions, single vacancies are mainly in the neutral charge state, such that vacancy recombination results in the formation of $V_2$ complexes with higher probability than the formation of a single NV centre within individual defect-clusters[31].

We further investigate the dynamics of vacancy diffusion by kinetic Monte Carlo (KMC) simulations at a temperature of 950 °C using the defect-cluster sizes simulated by MD as input. Our results indicate that approximately 30–40% of all single vacancies within individual defect-clusters are expected to build $V_2$ complexes or higher-order vacancy chains (see Supplementary Note 1). A nanometre-volume surrounding individual NV centres is calculated to host up to 4–6 $V_2$ complexes after low energy nitrogen implantation and thermal annealing. The exact configuration of $V_2$ spins depends, however, on the depth of individual NV centres (that is, ion channelling contribution). These complexes are known to be thermally stable[23] and are electron paramagnetic with a spin $I = 1$ (ref. 32). In fact, numerical evaluation of the interaction Hamiltonian representing NV–$V_2$ dipolar interactions shows that the presence of such a number of $V_2$ electron spins in the vicinity of an NV centre can be a dominating source of spin decoherence even in the close vicinity to the diamond surface, reducing $T_2$ times to a few μs[33] (see Supplementary Note 1). These vacancy complexes are hence predicated as the major cause of spin decoherence and

low formation yield of defect centres by nitrogen implantation. A method to suppress the formation of such defects during thermal annealing is thus a key step towards improving the quantum properties of spin defects.

**The $p^+$–i junction structure.** One way to suppress vacancy recombination is to charge single vacancies in the defect-clusters during thermal annealing. In this case, Coulomb repulsion for near-neighbour charged vacancies is on the order of $\sim$ eV and hence greatly reduces the formation probability of $V_2$ or higher-order vacancy complexes. To implement this concept, we propose a planar all-diamond structure comprising a thin diamond layer with high concentration of boron acceptors ($N_A \sim 10^{20}$ cm$^{-3}$) epitaxially grown on a ultra-pure diamond substrate with a low concentration of donors ($N_D \sim 10^{14}$ cm$^{-3}$), as sketched in Fig. 1a. At thermodynamic equilibrium, positive charge carriers (holes) diffuse from the boron-doped layer into the substrate, resulting in the formation of a space charge dipole at the interface, identical to a planar p–n junction in semiconductors. The interface between the boron-doped layer and the substrate presents a step-like change in the concentration of acceptors with $N_A/N_D$ ratio $>10^6$, generating a nanometre depletion zone in the boron-doped layer. As a result, the near-interface region of the substrate is positively charged by the free carriers.

The depth profile of holes is shown in Fig. 1c, extracted from a two-dimensional finite elements modelling (see Methods section). In the figure, the corresponding depletion layer is represented by the green-marked area. The space charge profile extends to depths of $>100$ nm in the substrate, even for a small thickness of the boron-doped layer of 6 nm. A suitable implantation energy of nitrogen thus localizes NV centres in this space charge layer, as seen in Fig. 1c. At a high concentration of holes in the substrate (Fermi level close to the valence band maximum), single vacancies are expected to be in the $2^+$ charge state during thermal annealing[34]. As a final step of sample preparation, the surface is etched for 7 nm to remove the boron-doped layer and leave a pristine and oxygen-terminated surface[35]. A reference area (no boron-doped layer, see Fig. 1a—right) is fabricated on the same diamond to exclude any substrate-related effects such as variation in the intrinsic concentration of impurities in our experiments.

**Formation yield of NV centres.** Figure 2a,b present examples of confocal microscopy scans from the reference and $p^+$–i junction (hereinafter referred to as 'sample') areas, respectively. The yield in the reference area is approximately 0.3, 0.6 and 1.0% for implantation energies of 2.5, 5.0 and 9.8 keV, respectively, which is in good agreement with literature[21]. The sample area shows a higher yield in comparison to the reference area for all three implantation energies, where an approximately twofold enhancement is observed, as seen in Fig. 2c. This supports the assumption that charging of vacancies at the $p^+$–i junction interface suppresses the formation of more complex implantation-induced defects during thermal annealing. Together with an enhanced self-diffusion at elevated temperatures[34], the charging effect leads to an increased number of single vacancies available for the formation of NV centres. Similar effects can also be responsible for the yield enhancement observed after the additional low-energy postimplantation electron irradiation before thermal annealing[36]. The reported enhancement factor of approximately 1.8 is similar to the values obtained in our experiments using the $p^+$–i junction structure. Charging of implantation defects may thus be a universal technique to tailor the evolution of implantation damages in the lattice of spin host materials.

The results presented in Fig. 2 also suggests a preferential path for the formation of NV centres: the implanted nitrogen atom is

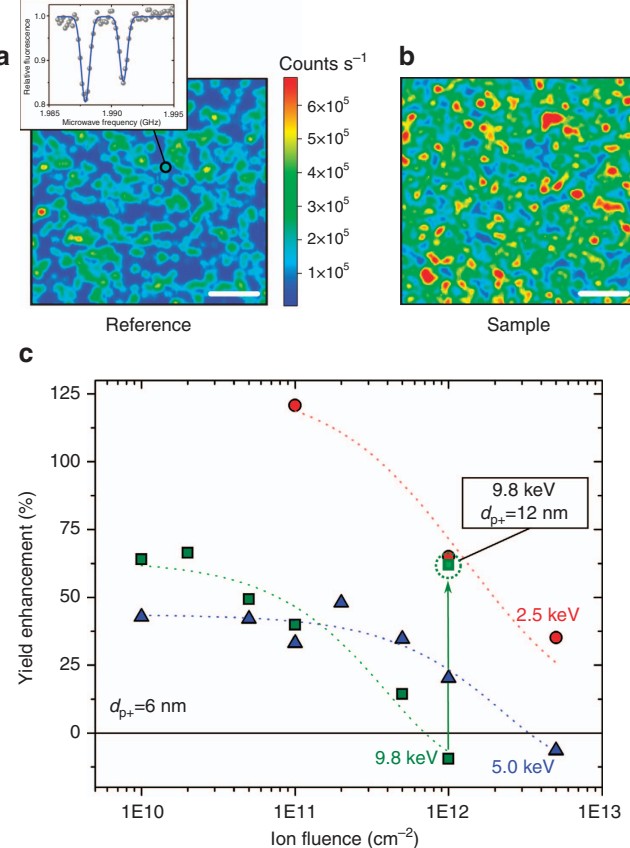

**Figure 2 | NV centres by nitrogen implantation across planar p⁺–i junction structure.** (**a,b**) Two-dimensional confocal maps of NV centres comparing the 'Reference' and 'Sample' areas, respectively, after ion implantation (5 keV $^{15}$N$^+$, $10^{10}$ cm$^{-2}$ fluence), thermal annealing and final etch (7 nm). The inset shows a typical optically detected magnetic resonance spectrum of the implantation-induced NV centres ($^{15}$N hyperfine splitting). Scale bars are 3 µm. (**c**) Enhancement in the formation yield of NV centres as a function of nitrogen ion fluence for three different implantation energies (2.5, 5.0 and 9.8 keV). The thickness of the boron-doped layer ($d_{p+}$) is 6 nm. The highlighted point corresponds to the yield enhancement for nitrogen implantation with 9.8 keV of energy in another diamond with a thicker ($d_{p+}$ = 12 nm) boron-doped layer (see text). The dashed-lines are guides to the eye. Horizontal error bars are negligible, whereas vertical error bars are in the range $< \pm 10\%$ (not shown for clarity).

located in a split-interstitial site surrounded by single vacancies that occupy the nearest neighbour sites[27,36]. MD simulations further reveal that as-implanted nitrogen atoms in diamond occupy interstitial rather than substitutional positions (60 and 40% calculated probabilities, respectively)[37]. Our results indicate that the recombination rate of a single vacancy and an interstitial nitrogen atom, followed by trapping a second vacancy located in the second-neighbour lattice position to form a stable NV centre is enhanced by vacancies being positively charged. Electrostatic repulsion between vacancies should thus not only prevent the formation of vacancy complexes but also facilitate the interstitial nitrogen atom (not charged) to occupy the nearest vacancy position.

From our results, we observe that the yield enhancement decreases with increasing nitrogen ion fluence for all implantation energies, as seen in Fig. 2c. This is due to the resulting charge compensation of holes induced by implantation-induced defects (for example, charge traps and donor impurities). As shown in

Fig. 2c, the ion fluence at which the enhancement is reduced to zero (critical ion fluence) shifts towards lower ion fluences for higher implantation energies. Since higher energies lead to a shift in the profile of implantation-induced defects towards the substrate region—as seen from the atomic profiles in Fig. 1c—we conclude that the charge compensation occurs in the substrate area. This is further supported by numerical simulations of the p⁺–i structure in the presence of compensating donors with different doping profiles (see Supplementary Note 2). Moreover, in the case of our structure, charge compensation of acceptor impurities directly in the boron-doped layer, as previously discussed in ref. 38, would result in the opposite behaviour regarding the critical ion fluence versus implantation energy.

In addition, the enhanced formation yield is still observed for the ion implantation with 9.8 keV of energy and $10^{12}$ cm$^{-2}$ ion fluence in the case of a second diamond fabricated with a thicker (12 nm) boron-doped layer (final etching of 17 nm; see the highlighted square in Fig. 2c and Supplementary Note 2). For this diamond, ion implantation with the same energy leads to a lower concentration of implantation-induced compensating defects in the substrate area, thus shifting the critical ion fluence towards higher values. This further supports that the charge compensation occurs in the substrate area rather in the boron-doped layer, as previously discussed. Importantly, the results obtained by the two different diamonds confirm indirectly the presence of holes in the substrate area of the fabricated structures.

**Spin properties of NV centres.** To address the impact of the surrounding implantation-induced defects, we use the NV centres themselves as sensors to probe the local environment. Comparing the sample and reference areas provides the opportunity to esti-mate directly the spin performance by using the p⁺–i junction structure. The experiments were carried out on defects created with 5.0 keV of implantation energy and $10^{10}$ cm$^{-2}$ ion fluence owing to the possibility to sample implantation sites with different contributions of ion channelling. The spin coherence $T_2$ (Hahn echo) and relaxation $T_1$ times (Fig. 3a,b, respectively) were measured for several NV centres at both regions. The depth of each NV centre was measured independently (see Methods section). As seen in Fig. 3a, $T_2$ times up to ∼180 µs are observed for NV centres confined within 2–8 nm of depth at the sample area.

By comparing NV centres with similar depths from the sample with the reference area, a tenfold improvement is seen for $T_2$ times for depths <5 nm in the sample area. This remarkable difference in $T_2$ times is also observed in the statistics of a larger number of NV centres in both areas (see Supplementary Fig. 7). Moreover, the enhancement in $T_2$ times is reproducibly observed for NV centres created using different configurations of the p⁺–i junction structure, for example, diamonds fabricated with different thicknesses of the boron-doped layer and different nitrogen implantation energies (see Supplementary Notes 2 and 3 for further information). Noticeably, $T_2$ times measured from NV centres in the reference area approach the values of those from the sample area with increasing depth (see Fig. 3a). This behaviour is attributed to the depth distribution of $V_2$ spins, which are the dominant noise source for NV centres at depths <5 nm in the reference region—as explained below. Nitrogen atoms implanted with 5.0 keV of energy are expected to be distributed as shown by the atomic profile in Fig. 3c by crystal-TRIM (CTRIM)[39]. In addition, the number of $V_2$ complexes surrounding NV centres within individual defect-clusters is expected to be in the range given by the orange-dashed lines in Fig. 3c. These values were obtained by KMC simulation using the MD simulation results as input (see Supplementary Note 1). As

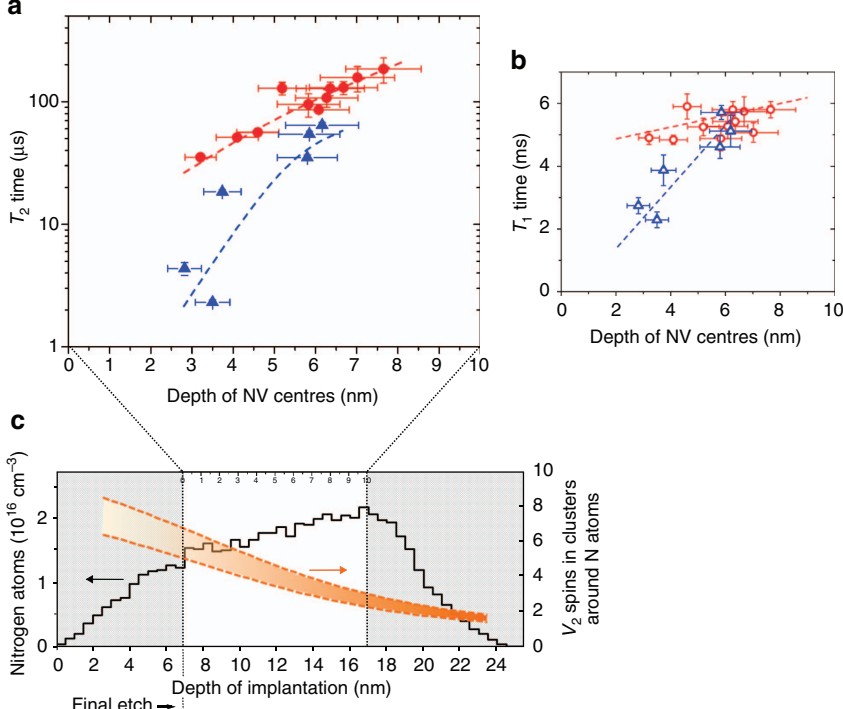

**Figure 3 | Enhanced spin properties of NV centres at the sample area.** (**a**,**b**) $T_2$ (Hahn echo) and $T_1$ times, respectively, of individual NV centres as a function of depth at the 'Sample' (circles) and 'Reference' (triangles) areas (magnetic field of $\sim 33$ mT aligned to the NV axis). The depth of each NV centre was measured independently by spin relaxation technique using gadolinium ($Gd^{3+}$) ions deposited on the diamond surface. Dashed-lines are guides to the eye. The error bars correspond to the s.d. of the data fit. (**c**) Depth profile of nitrogen atoms by CTRIM simulation for 5.0 keV implantation energy ($10^{10}$ cm$^{-2}$ ion fluence) in [100]-oriented diamond for an incident angle of 3°[30,35,37]. The two vertical dashed lines align the projected depth of the implanted nitrogen atoms to the experimental depth range of NV centres measured by spin relaxation technique, after the final etching of 7 nm. The orange lines delimit our simulated distribution profile of di-vacancies (30–40% of single vacancies are converted to di-vacancies within defect clusters, see text).

apparent from the figure, the number of $V_2$ complexes per defect-cluster is reduced by a factor of three within 10 nm of depth of NV centres (see upper scale limited by black-dashed vertical lines in Fig. 3c). In fact, for depths $>5$ nm, the number of $V_2$ spins in close vicinity of the NV centres becomes negligible. For this depth range, the magnetic noise from surface spins dominates the observed decoherence rate. Since the surface conditions are the same, similar values of $T_2$ are expected in the sample and reference areas, as indeed seen in our experiments (Fig. 3a).

However, NV centres located at depths $<5$ nm are surrounded by 4–6 $V_2$ electron spins (Fig. 3c). A comparison between the depth dependencies seen in Fig. 3a,c indicates that the additional noise contribution in the reference area arises mainly from the larger number of $V_2$ electron spins in the vicinity of NV centres. Furthermore, a similar behaviour is observed for the measured $T_1$ times, shown in Fig. 3b. The achieved values of 5–6 ms are significantly longer than those typically observed for near-surface NV centres[40–42], thus assuring the pristine quality of the processed diamond. Both $T_1$ and $T_2$ values and depth dependencies further support the assumption that our method suppresses the formation of $V_2$ complexes within individual defect-clusters.

**Noise spectroscopy.** To further analyse the magnetic noise affecting the NV centres at both the sample and reference areas, noise spectroscopy is utilized[43]. Dynamic decoupling sequences with associated, characteristic spectral filter functions[44,45] are used to sample the magnetic noise sensed by NV centres (same from Fig. 3a,b) in different frequency ranges. Coherence decays were measured using the Carr–Purcell–Meiboom–Gill

sequence (CPMG) with different numbers ($N$) of $\pi$-pulses and spectral decomposition technique[46,47] was used to extract the corresponding noise spectra $S(\omega)$.

Figure 4a shows the resulting noise spectra corresponding to two NV centres located at the two areas with similar depths ($\sim 3$ nm) extracted from CPMG-32 decoherence decays. Single-Lorentzian functions are used to fit the noise spectra extracted from NV centres in the sample area[46,47]. Figure 4b shows the related coupling strengths versus the depth of NV centres. The model used to fit the experimental data (dashed line) comprises two coupling strength contributions, $\Delta_1$ and $\Delta_2$. The first corresponds to the surface spin bath, with a $1/d^2$ depth dependency[40] (solid line). The second is a contribution with constant amplitude (well below the amplitude related to surface spins), which represents the coupling to residual paramagnetic spins in the diamond lattice such as P1 centres and $^{13}C$ nuclear spins.

Using this approach, a density of surface spins $\rho_{ss} \sim 10^{13}$ spins cm$^{-2}$ is estimated, which is in good agreement with previous experiments[40]. The derived value of $\rho_{ss}$ is precisely in the range of the spatial density of electronic states for oxygen-terminated diamond surfaces. Such states are responsible for the band bending at the bare diamond surface and junction barriers of corresponding Schottky diode structures[48]. Furthermore, noise spectra from NV centres in the sample area yield typical values of correlation times in the range of $\sim 10$ ns. For the mentioned density $\rho_{ss}$, the average distance of near-neighbour surface spins is approximately 1.3 nm, corresponding to a mutual coupling of $\sim 50$ MHz (see Supplementary Note 1). Hence, flip-flops among near-neighbour surface spins can lead to fluctuating magnetic noise at the NV centre location with correlation times in the range of $\sim 10$–50 ns. Therefore, our experimental correlation

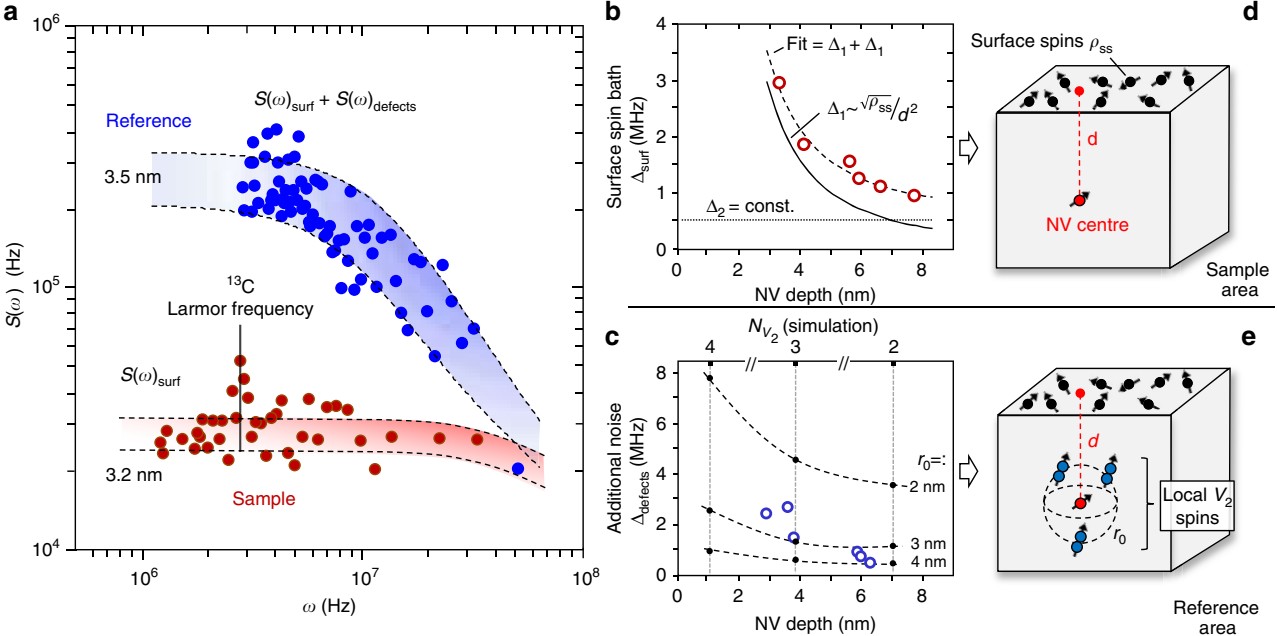

**Figure 4 | Suppressed magnetic noise for NV centres. (a)** Noise spectra sensed by two NV centres located at the 'Sample' and 'Reference' areas at similar depths ($\sim 3$ nm), as derived from coherence decay (CPMG pulse sequence, $N = 32$ $\pi$-pulses) by means of spectral decomposition technique. The dashed curves and shadowed areas establish margins for fitting the spectra by Lorentzian functions. **(b)** Extracted coupling strength as a function of depth for NV centres in the 'Sample' area. The dashed curve is a two-component fit comprising the model of surface spin bath[40] (solid curve (1)) and a constant noise contribution (dotted curve (2)). **(c)** The additional noise component (defects coupling strength) extracted from NV centres in the 'Reference' area is shown as blue dots. The black dots connected by the dashed curves represent the margins of the noise coupling strength calculated for different numbers of di-vacancy electrons spins. These margins were calculated assuming the simplified modelled NV-$V_2$ configuration depicted in panel **(e)**, which represents residual defects after nitrogen implantation and thermal annealing (see text and Supplementary Note 1 for details). **(d,e)** Schematic representations of the resulting spin systems around a single implantation-induced NV centre at the 'Sample' and 'Reference' areas, respectively.

times can be attributed mostly to spin–spin interactions between the near-surface NV centre and the surrounding surface spin bath.

As mentioned, the surfaces of both sample and reference areas were treated identically such that the two areas have identical surface spin bath characteristics. For this reason, we use a double-Lorentzian function to fit the noise spectra related to the reference region. This comprises the described fixed $S(\omega)_{surf}$ and an additional $S(\omega)_{defects}$ noise contributions, as depicted in Fig. 4a. For each depth of NV centres, the parameters for the $S(\omega)_{surf}$ were set fixed according to the experimental data in Fig. 4b. In this way, we extract independently the coupling strengths and correlation times related to the additional noise source, which is responsible for the faster decoherence decay of near-surface NV centres in the reference area seen in Fig. 3a.

For the additional $S(\omega)_{defects}$ noise contribution, the extracted correlation times are in the range of 70–100 ns. The coupling strengths versus the depth of NV centres are shown in Fig. 4c by blue circles. The same figure also shows margins for the coupling strength calculated for a modelled structure comprising a single NV centre surrounded by di-vacancy electron spins ($I = 1$). For simplicity, the di-vacancies are assumed to be equally distributed on the surface of a sphere of radius $r_0$ centred on the NV centre, as sketched in Fig. 4e. This allows us to reduce the variables involved in the calculations to $r_0$ and the number of di-vacancies only. Values for $r_0$ and the corresponding number of di-vacancies at different depths correlate to the results from the MD, KMC and CTRIM simulations (see Fig. 3c and Supplementary Note 1). Within the given margins, the calculated results are reasonably confined to the range obtained in our experiments for the additional magnetic noise component in the reference area (Fig. 4c). This allows us to attribute such noise to

the residual implantation-induced defects that are not fully eliminated by the thermal annealing in the reference area. Importantly, this noise signature of $V_2$ spins is only present in the reference region.

## Discussion

The discussed experimental and numerical modelling results in a consistent picture of the formation dynamics of NV centres by low-energy nitrogen implantation. They show that charging of single vacancies within defect-clusters is an effective method to suppress implantation-induced paramagnetic defects that degrade the spin performance of near-surface NV centres. This aspect is of particular interest since it can be extended to the broader field of ion implantation in semiconductor materials. Often, structural defects significantly deteriorate the optical, spin and charge transport properties, especially in the vicinity to surfaces and interfaces[49]. For diamond, we identify the formation of $V_2$ complexes as a potential dominant source of spin decoherence, whereas for other materials different structural defects may account for spin decoherence. In all cases, there is a critical threshold for the size of defect-clusters that may not be eliminated by thermal annealing[50]. As a result, charging of implantation-induced damage may be an universal tool towards better quantum properties of implanted spin defects.

In the future, the concept of defect charging may be implemented into various methods beyond the suggested $p^+$–i junction structure and also combined with different postimplantation techniques such as ion co-implantation[51,52], possibly increasing the yield of NV centres near the diamond surface even further. In addition, the presented results bring new insights into the dynamic mechanisms of self-diffusion and

defect formation at single implantation sites, which enables new methods to further improve the engineering of spin defects. Implantation of multiple atoms and molecules[53] or implantation into more complex heterostructures combining strain engineering[54] with the presented structure might allow the fabrication of complex defect arrays. Extending the coherence times of near-surface spins and increasing the formation yield are decisive steps towards more sensitive nanoscale quantum sensing and coupling of single spins to, for example, superconducting resonators.

## Methods

**Numerical simulations.** MD simulations were used to investigate the implantation of nitrogen atoms into the diamond lattice. The atomic interactions are represented by a combination of a bond-ordered Tersoff and a two-body Ziegler–Biersack–Littmark (for small atomic separation) potentials. The implanted atom is placed at a distance from the diamond surface beyond the Tersoff potential action range, with a kinetic energy of 4.0 keV at nominally zero-angle ($z$ axis). Furthermore, several events of implantation were simulated separately for different $x$–$y$ positions of the nitrogen atom in order to investigate the ion channelling effect.

The annealing process was simulated by means of KMC based on a model of hopping frequency of defects $\Gamma = \omega_a e^{\frac{E_a}{k_B T}}$ at a temperature of 950 °C. It includes the most important loss mechanisms for vacancies, such as migration to the surface, recombination with interstitials and the formation of di-vacancies or higher-order vacancy complexes. The initial distribution of vacancies and nitrogen atom after implantation are taken from the results of MD simulations.

The decoherence caused by di-vacancies on the electronic spin of single NV centres is estimated by numerical calculations. We model $N_{V2}$ di-vacancies surrounding a central NV centre with varying distance–radius $r_0$. The di-vacancy electron spins ($I = 1$) are coupled to each other and to the central NV centre electron spin ($S$) via magnetic dipole–dipole interaction. The corresponding equations are solved numerically to obtain the $T_2$ Hahn echo and CPMG-$N$ decoherence decays and corresponding noise coupling strength from the central NV centre electron spin.

The one-dimensional depth profile of charge carriers presented in Fig. 1c was taken from a two-dimensional simulation of the thermodynamics equilibrium of the described planar $p^+$–i junction structure by means of finite elements method (SILVACO software, available at http://www.silvaco.com/). Standard diamond parameters were used such as carrier concentration and mobility and the simulation mesh was optimized taking into account the relevant length scales. The simulated profile exhibits a quadratic-hyperbolical depth dependency of the concentration in the form $p(z) \sim 1/(1 + z/L_D)^2$, where $L_D$ is the characteristic Debye length related to the given boron doping in the $p^+$ layer. The charge contribution of the ionized nitrogen donors in this subsurface region of the substrate can be neglected since $p(z) \gg N_D$.

Further details about the described simulation methods can be found in Supplementary Note 1.

**Boron-doped diamond growth and creation of NV centres.** The heavily boron-doped diamond layers for the $p^+$–i junction structures were overgrown by microwave plasma chemical vapour deposition in an *ASTEX* 5,000 reactor on the surface of a [100]-oriented single-crystal electronic-grade diamond substrates (<1 p.p.b. nitrogen and boron, 1.1% $^{13}$C—Element Six). The as-polished surfaces had, before growth, a root mean square roughness of <1 nm (measured by atomic force microscopy). The boron doping was realized *in situ* by introducing a solid boron rod in the plasma. Such a solid-state doping method enables the diamond overgrowth with a very sharp doping profile, approaching $10^{20}$ cm$^{-3}$ nm$^{-1}$. The used process parameters are described elsewhere[55]. Importantly, due to absence of impurity diffusion in the diamond lattice, these layers remained stable at the used annealing conditions.

NV centres were created by implanting $^{15}$N$^+$ ions with energies of 2.5, 5.0 and 9.8 keV by means of a focussed ion beam in a home-built setup. The diamonds were annealed at 950 °C under high vacuum (<10$^{-6}$ mbar) for approximately 2 h. The fabrication of the reference area and final etching steps were performed using oxygen inductively coupled plasma. This process enables nanometre-precise etching rate and was demonstrated to yield a pristine, oxygen-terminated surface while still preserving the optical and spin properties of NV centres located at a few nanometres of depth[35].

**Depth calibration.** The depth of individual NV centres at both the sample and reference areas were calibrated by means of spin relaxation technique[56]. Reference values of $T_1$ were measured for the oxygen-terminated surface. Afterwards, a film (~1 μm in thickness) containing Gd$^{3+}$ ions at a nominal concentration of 1 M was spin-coated on the diamond surface and $T_1$ times were measured to extract the gadolinium-induced spin relaxation. The depth was estimated using the model previously described in ref. 56.

**Spin properties of NV centres.** For all spin measurements, a magnetic field of ~33 mT was applied along one of the possible orientations of NV centres. $T_2$ times were measured using a Hahn-echo sequence and the transverse relaxation times in the rotating frame ($T_{2\rho}$) were measured using a CPMG dynamic decoupling sequence with $2 < N < 256$ $\pi$-pulses. These were fitted using:

$$C(t) = A \cdot \exp\left[-\left(\frac{t}{T_2}\right)^n\right] \sum_{i=0}^{N_R} \exp\left[-\left(\frac{t - i\tau_r}{2\tau_c}\right)^2\right] + C \quad (1)$$

where the first exponential term represents the main decoherence decay envelope with corresponding $T_2/T_{2\rho}$ time ($1 \le n \le 3$), and the second sum term represents the electron spin echo envelope modulation due to $^{13}$C spins in the diamond lattice[22]. $N_R$ is the number of revivals, $\tau_r$ is the corresponding periodicity and $\tau_c$ is the initial fast coherence decay related to the electron spin echo envelope modulation. $T_1$ measurements were fitted using a single exponential in the form $C(t) = A \cdot \exp\left[-\frac{t}{T_1}\right] + C$.

**Noise spectra.** The decoherence decays ($C(t) = \exp[-\chi(t)]$) measured using CPMG pulse sequence were used to extract the noise spectra from individual NV centres. Noise spectral decomposition based on the filter function ($F_t(\omega)$) arising from the used pulse sequence[44,46] was used to deconvolve the noise spectra following:

$$\chi(t) = \frac{1}{\pi} \int_0^\infty S(\omega) \frac{F_t(\omega)}{\omega^2} d\omega \quad (2)$$

The noise spectra presented are fitted according to:

$$S(\omega) = \sum_i \frac{\Delta_i^2 \tau_i}{\pi} \frac{1}{1 + (\omega \tau_i)^2} \quad (3)$$

where $i = 1$ and 2 represents a single- or double-Lorentzian fit, respectively, $\Delta$ is the noise coupling strength and $\tau$ is the correlation time between the NV centre and the spin bath.

**Data availability.** All relevant data are available upon request from the corresponding authors.

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

## Acknowledgements

The authors acknowledge the financial support from SFB/TR 21 as well as the Volkswagen Foundation and the Max Planck Society. D.A. acknowledges the financial support by the DFG via the SFB 716. F.F.d.O. acknowledges CNPq for the financial support through the project No. 204246/2013-0. We thank P. Deák for fruitful discussions.

## Author contributions

A.D. designed the concept idea; F.F.d.O. and A.D. performed the experiments and data analysis, assisted by S.A.M.; D.A. and T.H. performed the MD and KMC simulations; Y.W. performed the numerical calculations of the NV–$V_2$ interaction Hamiltonian; A.P. assisted with the boron-doped diamond growth; P.N., A.D. and J.W. supervised the work; all authors contributed to the manuscript writing.

## Additional information

**Competing interests:** The authors declare no competing financial interests.

**Publisher's note**: 

