## [Peer review File · Nature Communications]

Reviewer #1 (Remarks to the Author):

In their article de Oliveira et al present results from an integrated experimental and simulation study of the formation and spin properties of NV- centers in nitrogen implanted and thermally annealed diamond.

In my opinion this work presents a major breakthrough in this topic area.

NV- centers are popular because of a series of properties that enable sensitive magnetometry and stimulate ideas for spin-photon qubits. Ion implantation followed by thermal annealing has been used for over decade to form NV-centers. But yields, NV/N, are notoriously low and spin properties are inferior vs. NV- centers formed during growth.

Much has been tried, such as co-implantation, exposure to electrons, higher annealing temperatures and pressure, etc. and there have been great advances in simulations of NV-formation kinetics.

The present work demonstrates convincingly that control of the charge state of implanted associated defects (vacancies) a) enhances the formation of NV- centers, b) improves the T2 of NV-centers and c) reduces magnetic noise. This is achieved through clever implementation of a boron doped layer with the NV- center containing layers in diamond. The simulations of vacancy formation and basic electrostatics combined with damage formation are convincing. The results are very encouraging, up to 2x more NV-centers for low energy implantation (where the absolute yield is very low to begin with), up to 10x better coherence time, and reduced noise. The methodology could also lead to further improvements when combined with other levers (e. g. co-implantation, hot implantation, structured substrates, etc.).

I recommend publication in Nat. Comm.

I also like to suggest the following minor points:

1. I suggest that the authors clearly state what the estimated total NV- yields are here, so that the 2x yield enhancement can be put into perspective. Please also comment on the relative yield enhancements reported by other means (vs. plain thermal annealing), e. g. higher temperature annealing, co-implantation, etc. (some of which some of the present authors have reported on earlier). I believe that this would help the readers calibrate the relative importance of the yield enhancements reported here.
2. Please comment on the NV- to NV0 ratio for these device conditions.

Reviewer #2 (Remarks to the Author):

The authors investigate the formation of NV centres by means of low energy ion implantation, by means of experiments supported by molecular dynamics simulations and simple models of the decoherence. There is currently enormous interest in NV centres given their importance in spin sensing and for potential quantum information applications. Within this context there is also enormous interest in techniques to place NV centres in reasonably accurately engineered structures and to do so without unduly compromising on their very valuable coherence and fluorescence properties. A key problem, apart from low yield, is the degradation of the properties of near surface NV centres. Ion implantation has especial drawbacks in loss of coherence due to implantation-induced damage to the lattice.

The manuscript presents and tests an interesting conjecture that implantation of nitrogen atoms into a charged layer, generated by a surface boron-doped layer suppresses the formation of V2 di-vacancies which are a key cause of the degradation of near-surface NV centres. A set of

experimental studies are presented which compare the behaviour of a "sample area" with the charged vacancies with a "reference" sample.

The authors conclude that Coulomb repulsion effects suppress V2 formation to a sufficient extent to allow the formation of NV centers with T2 times of order ~ 0.1 ms. While not as favourable as the \sim ms coherence times of deeper NV centers this would still represent an important improvement. Evidence is presented that even the yield is improved by charging. On those grounds the paper deserves to be published.

There are one or two weaknesses in my view which need to be addressed:

1) A central (maybe "the" central) result is Fig 3a (plus inset) where a comparison between the coherence times (T₂ and T₁) for sample and reference are compared (I note that the figure does not explicitly state red circles are sample, blue triangles are reference sample but I infer this: please fix that).

My key concern is that if one takes out the two blue triangles at 3nm the reference fit (the dashed line) would be quite different; in particular, there would be far less contrast between the sample and reference data.

The problem here is that there are a lot of data points at larger depths, where there is little contrast but only one or two data points at shallow depths of 2-3nm where the order-of-magnitude difference is claimed. Is it possible that these two lowest points (in each of red/blue sets) are atypical?

2) It is argued that the origin of the short, microsecond T₂ coherence times of the reference is the effect of the V2 spins. A simulation of the coherence dynamics is presented in the Suppinfo. The model seems a little suspect to me. It is stated that the surrounding V2 are in a sphere of radius r₀. As I infer from the illustrations and context, they are in fact on the *surface* of a sphere of radius r₀.

This would seem unphysical. Have the authors excluded the possibility of equivalent sites, where the V2 could become degenerate and flip-flop with each other? Is this assumption of fixed r₀ necessary? One would expect a random distribution around the expected mean r₀. With calculations using N=3-5 V2 one would expect order-of-magnitude fluctuations in T₂ in any case on statistical grounds.

Please note that the same capital N is used for pulse number as well as number of neighbouring V2 spins. This is potentially confusing.

ALSO:

Some minor typos/confusing grammar

Page 1 Col1 "retraining" should be retaining

Page 1 col 2 "degraded properties ..of the spin..restrain their interaction time"

You mean limit or restrict their interaction time.

P 1 two confusing instances of the term ranging. Eg do you have a set of values spanning 5ms (eg "T1 ranging from from 1-6 ms" , do you mean improvements ie changes in T1 ranging from 1 to 5 ms?. Ditto spatial resolution ranging 5-10nm.

Do you mean ranging from 5 to 10 nm?

Caption of Fig 1 is confusing- the sentence stating " sacrificial boron-doped layer...- region named "Sample area". Maybe clarify that the boron layer is not the sample area.

In summary: Although some aspects in terms of justifying attribution of most effects to V2 as well as the magnitude of the difference are not 100% persuasive, on balance I find the study interesting and intriguing -and likely to stimulate further interest- so worth the risk of publishing even if the conjecture that charging gives order of magnitude technical improvement is later found to be partially flawed or of limited usefulness.

Reviewer #3 (Remarks to the Author):

In their manuscript, the authors discuss the microscopic formation dynamics of the nitrogen-vacancy (NV) centers created via low-energy ion implantation followed by thermal annealing, based on molecular dynamics and kinetic Monte-Carlo simulations. The authors then propose and experimentally demonstrate a novel method using a p+-i junction structure to suppress the formation of unwanted di-vacancies around the nitrogen atom. T_1 , T_2 , and the spin noise spectrum of the created NV center spins are carefully measured and analyzed. These experimental results corroborate the above-mentioned numerical simulations.

I recommend this manuscript for publication in Nature Communications. The present work combines sophisticated numerical tools with the state-of-the-art and diverse experimental techniques, succeeding in providing a clear and consistent picture on the formation dynamics of the NV centers, which is surely of great importance in the NV-research community. I have only a few comments/clarifications. Otherwise the manuscript is clearly written.

The title is catchy, but is not informative. I would suggest expanding the title such as "Tailoring spin defects in diamond by ..." to include more information after "by".

In Figs. 3(a) and (b), legends or caption information regarding which data are for "sample" and which for "reference" are missing; specify which is which. Also clarify not only the implantation energy (5 keV) but also the ion fluence used to create the NV centers measured in Figs. 3(a) and (b). Because Fig. 2(c) tells that the yield enhancement is dependent both on the implantation energy and the ion fluence, they are equally important information. Likewise, specify the ion fluence for the CTRIM simulation in Fig. 3(c).

Below is a list of minor typographical errors I have noticed.

- (1) In line 15, "nitrogen vacancy" should read "nitrogen-vacancy".
- (2) In line 18, "diamond doped structure" should read, e.g., "boron-doped diamond structure".
- (3) In line 47, "retraining" should read "retaining".
- (4) In line 65, "road block for" should read "roadblock to".
- (5) In line 115, "are show" should read "are shown".
- (6) In lines 135-6, "substrates ... results" should read "substrates ... result".
- (7) In lines 172-3, "near-neighbors" should read "near-neighbor".
- (8) In line 213, "referred as" should read "referred to as".
- (9) In the caption of Fig. 2, " ^{15}N hyperfine splitting" should read " ^{15}N hyperfine splitting".
- (10) In lines 544 and 546, "therm" should read "term".
- (11) In reference [4], the eighth author should be "A. S. Sørensen".
- (12) In reference [7], the fifth author should be "C. T. Schulte-Herbrüggen", and the journal volume and page should be "506" and "204", respectively.
- (13) In reference [12], the sixth author should be "L. C. L. Hollenberg".
- (14) In reference [44], the article number should be "858".
- (15) In reference [53], the article number should be "1607".
- (16) In line 6, page 2 of SI, "two body" should read "two-body".
- (17) In line 7, page 4 of SI, "therms" should read "terms".
- (18) In line 23, page 5 of SI, "a N di-vacancies" should read "N di-vacancies".

- (19) In line 3, page 7 of SI, "is show" should read "is shown".
- (20) In line 2, page 14 of SI, "an NV centers" should read "an NV center".
- (21) In line 3, page 15 of SI, "are show" should read "are shown".
- (22) In line 3, page 16 of SI, "P\$_1\$ centers" should read "P1 centers".

Reply to the comments of the reviewers

Reviewer 1:

Recommendation: "I recommend publication in Nat. Comm."

Comments:

In their article de Oliveira et al present results from an integrated experimental and simulation study of the formation and spin properties of NV- centers in nitrogen implanted and thermally annealed diamond.

In my opinion this work presents a major breakthrough in this topic area.

NV- centers are popular because of a series of properties that enable sensitive magnetometry and stimulate ideas for spin-photon qubits. Ion implantation followed by thermal annealing has been used for over decade to form NV-centers. But yields, NV/N, are notoriously low and spin properties are inferior vs. NV- centers formed during growth.

Much has been tried, such as co-implantation, exposure to electrons, higher annealing temperatures and pressure, etc. and there have been great advances in simulations of NV-formation kinetics.

The present work demonstrates convincingly that control of the charge state of implanted associated defects (vacancies) a) enhances the formation of NV- centers, b) improves the T2 of NV-centers and c) reduces magnetic noise. This is achieved through clever implementation of a boron doped layer with the NV- center containing layers in diamond. The simulations of vacancy formation and basic electrostatics combined with damage formation are convincing. The results are very encouraging, up to 2x more NV-centers for low energy implantation (where the absolute yield is very low to begin with), up to 10x better coherence time, and reduced noise. The methodology could also lead to further improvements when combined with other levers (e. g. co-implantation, hot implantation, structured substrates, etc.).

I also like to suggest the following minor points:

1. I suggest that the authors clearly state what the estimated total NV- yields are here, so that the 2x yield enhancement can be put into perspective.

Answer:

To address the raised concern we have added the following sentence to the section "Formation yield of NV centers", line 213: "*The yield in the reference area is approximately 0.3, 0.6 and 1.0 percent for implantation energies of 2.5, 5.0 and 9.8 keV, respectively, which is in good agreement with literature[21].*".

In addition, to enhance the two-fold yield enhancement as suggested by the reviewer, the sentence "*an increase of approximately 100%*" in line 216 was replaced by "*an approximately two-fold enhancement*".

Please also comment on the relative yield enhancements reported by other means (vs. plain thermal annealing), e. g. higher temperature annealing, co-implantation, etc. (some of which some of the present authors have reported on earlier). I believe that this would help the readers calibrate the relative importance of the yield enhancements reported here.

Answer:

We agree that a comparison with other methods would help the community to measure the relative importance of the improvement reported in this work. The reported yield enhancement of 1.8 by low-energy electron irradiation (J. Schwartz *et al*, NJP 14, 043024 (2012)) in comparison to plain thermal annealing is very similar to the value obtained in our work. Since they have used irradiation energies well below the threshold for vacancy creation, we believe that similar charging effects may also have played an important role in their case. In this sense, the following has been added to the manuscript in line 221: "*Similar effects can also be responsible for the yield enhancement observed after the*

additional low-energy post-implantation electron irradiation before thermal annealing[35]. The reported enhancement factor of approximately 1.8 is similar to the values obtained in our experiments using the p⁺-i junction structure. Charging of implantation defects may thus be a universal technique to tailor the evolution of implantation damages in the lattice of spin host materials. (new paragraph) The results presented in figure 2 also suggests a preferential path for the formation of NV centers: ...”

Recently, we have started to investigate the combined use of the p⁺-i junction and ion co-implantation. Results presented in F. Fávaro de Oliveira *et al*, Phys. Status Solidi A, 1–7 (2016) indicates that vacancy recombination (i.e. formation of vacancy complexes) during thermal annealing are responsible for the inefficient creation of additional NV centers by the co-implantation induced vacancies. In this regard, we limit ourselves to the addition of the following sentences to the “Discussion” section of the manuscript: *(new paragraph) In the future, the concept of defect charging may be implemented into various methods beyond the suggested p⁺-i junction structure and also combined with different post-implantation techniques such as ion co-implantation[49, 50], possibly increasing the yield of NV centers near the diamond surface even further.”.*

2. Please comment on the NV⁻ to NV⁰ ratio for these device conditions.

Answer:

The exact NV⁻/NV⁰ ratios for the NV centers and the samples used in this work were not measured. For the case of 5.0 keV implantation energy under the etched surface (another sample, parameters similar to figure 3), fitting the photo-luminescence spectra acquired from single NV centers at 532 nm excitation using NV⁻ and NV⁰ reference spectra indicates that the NV⁻/NV⁰ ratio may be already in the range between 50 - 90% in the reference area. This can be attributed to the high quality of the diamond surface after etching.

Reviewer 2:

Recommendation: “the paper deserves to be published”

Comments:

The authors investigate the formation of NV centres by means of low energy ion implantation, by means of experiments supported by molecular dynamics simulations and simple models of the decoherence. There is currently enormous interest in NV centres given their importance in spin sensing and for potential quantum information applications. Within this context there is also enormous interest in techniques to place NV centres in reasonably accurately engineered structures and to do so without unduly compromising on their very valuable coherence and fluorescence properties. A key problem, apart from low yield, is the degradation of the properties of near surface NV centres. Ion implantation has especial drawbacks in loss of coherence due to implantation-induced damage to the lattice.

The manuscript presents and tests an interesting conjecture that implantation of nitrogen atoms into a charged layer, generated by a surface boron-doped layer suppresses the formation of V2 di-vacancies which are a key cause of the degradation of near-surface NV centres. A set of experimental studies are presented which compare the behaviour of a “sample area” with the charged vacancies with a “reference” sample.

The authors conclude that Coulomb repulsion effects suppress V2 formation to a sufficient extent to allow the formation of NV centers with T2 times of order ~0.1 ms . While not as favourable as the ~ms coherence times of deeper NV centers this would still represent an important improvement. Evidence is presented that even the yield is improved by charging. On those grounds the paper deserves to be published.

There are one or two weaknesses in my view which need to be addressed:

1) A central (maybe “the” central) result is Fig 3a (plus inset) where a comparison between the coherence times (T₂ and T₁) for sample and reference are compared (I note that the figure does not explicitly state red circles are sample, blue triangles are reference sample but I infer this: please fix that).

Answer:

The correct labels have been added to the capture of figure 3.

My key concern is that if one takes out the two blue triangles at 3nm the reference fit (the dashed line) would be quite different; in particular, there would be far less contrast between the sample and reference data. The problem here is that there are a lot of data points at larger depths, where there is little contrast but only one or two data points at shallow depths of 2-3nm where the order-of-magnitude difference is claimed. Is it possible that these two lowest points (in each of red/blue sets) are atypical?

Answer:

First of all, we would like to reinforce that all the results in figure 3 were produced using two areas (“sample” and “reference”) of the **same** diamond. As mentioned in lines 205-209, the reference **area** is fabricated on the same diamond substrate for the reasons given in the manuscript. Regarding the raised question, we measured initially ~30 NV centers on the bare oxygen-terminated diamond surface in each region of the particular diamond used to produce the data shown in Fig. 3. Short T₂ times (of a few microseconds) in the reference area are not atypical and, in fact, the average T₂ times from NV centers in the “sample” area is much higher than in the “reference” area. This tendency has been observed in different substrates fabricated using different conditions, e.g. different thicknesses of the boron-doped area, different implantation energies and ion fluences.

To address this question in more details, a section has been added to the Supplementary Information material with the relevant data in Fig SI-7. Furthermore, the link to this section in the main text (discussion of figure 3) has been added as “*This remarkable difference in T₂ times is also observed in the statistics of a larger number of NV centers in both areas. Moreover, the enhancement in T₂ times is reproducibly observed for NV centers created using different configurations of the p⁺-i junction*”

structure, e.g. diamonds fabricated with different thicknesses of the boron-doped layer and different nitrogen implantation energies (see SI for further information).”.

2) It is argued that the origin of the short, microsecond T_2 coherence times of the reference is the effect of the V2 spins. A simulation of the coherence dynamics is presented in the Suppinfo. The model seems a little suspect to me. It is stated that the surrounding V2 are in a sphere of radius r_0 . As I infer from the illustrations and context, they are in fact on the *surface* of a sphere of radius r_0 .

This would seem unphysical. Have the authors excluded the possibility of equivalent sites, where the V2 could become degenerate and flip-flop with each other? Is this assumption of fixed r_0 necessary? One would expect a random distribution around the expected mean r_0 . With calculations using $N=3-5$ V2 one would expect order-of-magnitude fluctuations in T_2 in any case on statistical grounds.

Answer:

The presented model is a simple estimation on the effect of the presence of a few electron spins in the vicinity of an NV center. This calculation was used to highlight that a few electron spins can be indeed the dominant source of decoherence, even in the close vicinity of the diamond surface. This feature plays a major role in the spin coherence times of near-surface NV centers created by nitrogen implantation, but has not been explored experimentally up to now due to the lack of a technique that allows a direct evaluation. In fact, the strong spin decoherence and magnetic noise spectra from NV centers in the reference region (standard nitrogen implantation) can be well explained by this attribution.

We deliberately used the simplified distribution of di-vacancies on the surface of a sphere with radius r_0 for the following reasons: first, figure 3 does not provide large enough statistics on spin decoherence for us to use the experimental results to numerically model the di-vacancy configuration in a more precise way. Second, several rounds of the MD and KMC simulations would have to be performed to extract the possible configurations of defect-clusters and consequently di-vacancies around NV centers. This would be extremely time-consuming. Such a high-level of details in simulations and numerical calculations would be above the scope of this work.

The statements made above regarding mutual spin flip-flops and random distribution of divacancies are correct, but the purpose of the presented calculations was not to model exactly these features and correlate them to the experimental results. Instead, we aimed to estimate *margins* where the NV spin decoherence and magnetic noise spectra correlate with the presence of a certain number of electron spins nearby. These margins are presented in figure 4 c) as the black-dotted lines. We would like to emphasize that nowhere in the main text or Supplementary Information an *exact correlation* between a modelled configuration of di-vacancies and experimental results was claimed. Nevertheless, to address this issue, the text has been modified attempting to state this more clearly (see the list of modifications below).

Please note that the same capital N is used for pulse number as well as number of neighbouring V2 spins. This is potentially confusing.

Answer:

To address this issue, the number of di-vacancies is represented by N_{V2} through the entire manuscript and Supplementary Information.

ALSO:

Some minor typos/confusing grammar

Page 1 Col1 “retraining” should be retaining

Page 1 col 2 “degraded properties ..of the spin..restrain their interaction time”

You mean limit or restrict their interaction time.

P 1 two confusing instances of the term ranging. Eg do you have a set of values spanning 5ms (eg “T1 ranging from from 1-6 ms” , do you mean improvements ie changes in T1 ranging from 1 to 5 ms?. Ditto spatial resolution ranging 5-10nm.
Do you mean ranging from 5 to 10 nm?

Caption of Fig 1 is confusing- the sentence stating “ sacrificial boron-doped layer...- region named “Sample area”. Maybe clarify that the boron layer is not the sample area.

Answer:

These mistakes have been corrected and the text has been modified accordingly.

In summary: Although some aspects in terms of justifying attribution of most effects to V2 as well as the magnitude of the difference are not 100% persuasive, on balance I find the study interesting and intriguing - and likely to stimulate further interest- so worth the risk of publishing even if the conjecture that charging gives order of magnitude technical improvement is later found to be partially flawed or of limited usefulness.

Reviewer 3:

Recommendation: "recommend this manuscript for publication in Nature Communications"

Comments:

In their manuscript, the authors discuss the microscopic formation dynamics of the nitrogen-vacancy (NV) centers created via low-energy ion implantation followed by thermal annealing, based on molecular dynamics and kinetic Monte-Carlo simulations. The authors then propose and experimentally demonstrate a novel method using a p+i junction structure to suppress the formation of unwanted di-vacancies around the nitrogen atom. T₁, T₂, and the spin noise spectrum of the created NV center spins are carefully measured and analyzed. These experimental results corroborate the above-mentioned numerical simulations.

I recommend this manuscript for publication in Nature Communications. The present work combines sophisticated numerical tools with the state-of-the-art and diverse experimental techniques, succeeding in providing a clear and consistent picture on the formation dynamics of the NV centers, which is surely of great importance in the NV-research community. I have only a few comments/clarifications. Otherwise the manuscript is clearly written.

The title is catchy, but is not informative. I would suggest expanding the title such as "Tailoring spin defects in diamond by ..." to include more information after "by".

Answer:

The title has been modified to "*Tailoring spin defects in diamond by lattice charging*".

In Figs. 3(a) and (b), legends or caption information regarding which data are for "sample" and which for "reference" are missing; specify which is which.

Answer:

This issue has been addressed, in response to reviewer 2.

Also clarify not only the implantation energy (5 keV) but also the ion fluence used to create the NV centers measured in Figs. 3(a) and (b). Because Fig. 2(c) tells that the yield enhancement is dependent both on the implantation energy and the ion fluence, they are equally important information. Likewise, specify the ion fluence for the CTRIM simulation in Fig. 3(c).

Answer:

The ion fluence has been added to the text and capture of figure 3.

Below is a list of minor typographical errors I have noticed.

- (1) In line 15, "nitrogen vacancy" should read "nitrogen-vacancy".
- (2) In line 18, "diamond doped structure" should read, e.g., "boron-doped diamond structure".
- (3) In line 47, "retraining" should read "retaining".
- (4) In line 65, "road block for" should read "roadblock to".
- (5) In line 115, "are show" should read "are shown".
- (6) In lines 135-6, "substrates ... results" should read "substrates ... result".
- (7) In lines 172-3, "near-neighbors" should read "near-neighbor".
- (8) In line 213, "referred as" should read "referred to as".
- (9) In the caption of Fig. 2, " ^{15}N hyperfine splitting" should read " ^{15}N hyperfine splitting".
- (10) In lines 544 and 546, "therm" should read "term".
- (11) In reference [4], the eighth author should be "A. S. Sørensen".
- (12) In reference [7], the fifth author should be "C. T. Schulte-Herbrüggen", and the journal volume and page should be "506" and "204", respectively.
- (13) In reference [12], the sixth author should be "L. C. L. Hollenberg".

- (14) In reference [44], the article number should be "858".
- (15) In reference [53], the article number should be "1607".
- (16) In line 6, page 2 of SI, "two body" should read "two-body".
- (17) In line 7, page 4 of SI, "therms" should read "terms".
- (18) In line 23, page 5 of SI, "a N di-vacancies" should read "N di-vacancies".
- (19) In line 3, page 7 of SI, "is show" should read "is shown".
- (20) In line 2, page 14 of SI, "an NV centers" should read "an NV center".
- (21) In line 3, page 15 of SI, "are show" should read "are shown".
- (22) In line 3, page 16 of SI, "P\$_1\$ centers" should read "P1 centers".

Answer:

We highly acknowledge the level of details given in this list. The pointed issues were corrected.

Changes applied to the main text:

The strikethrough text is the original version; the red text has been corrected or added

- I. The title of the manuscript has been changed to “*Tailoring spin defects in diamond by charging effects*”.
- II. Line 16: “*nitrogen vacancy*” has been changed to “*nitrogen-vacancy*”.
- III. Line 18: “*diamond doped*” has been changed to “*boron-doped diamond*”.
- IV. Line 47: “*retraining*” has been corrected to “*retaining*”.
- V. Line 52: “*ranging*” has been changed to “*in the range of*”.
- VI. Line 63: “*restrain*” has been changed to “*limit*”.
- VII. Line 65: “*road block for*” has been corrected to “*roadblock to*”.
- VIII. Line 88-89: “*ranging*” has been removed.
- IX. Line 115: “*show*” has been corrected to “*shown*”.
- X. Line 116: “*contribution*” has been changed to “*presence*”.
- XI. Line 136: “*results*” has been corrected to “*result*”.
- XII. Line 172-173: “*near-neighbors*” has been corrected to “*near-neighbor*”.
- XIII. Lines 178, 180 and 187: subscripts are now in text format (not italic).
- XIV. Capture of figure 1: “*the final fabricated region is*” has been added.
- XV. Capture of figure 2 and lines 207, 260, 263, 516: to avoid any misinterpretation regarding the definition of a sample (diamond substrate) and the “sample” area (p^+ -i junction structure after final etching), “*sample*” now reads “*diamond*”.
- XVI. Line 213: “*to*” has been added.
- XVII. Line 214: the following text has been added: “*The yield in the reference area is approximately 0.3, 0.6 and 1.0 percent for implantation energies of 2.5, 5.0 and 9.8 keV, respectively, which is in good agreement with literature[21].*”.
- XVIII. Line 216: “*an increase of approximately 100%*” has been changed to “*an approximately two-fold enhancement*”.
- XIX. Line 217: for clarity, the text has been modified from “*This supports the assumption that positively-charged vacancies localized within the space charge layer lead to enhanced self-diffusion[32] and an increased number of single vacancies available for the formation of NV centers.*” to “*This supports the assumption that charging of vacancies at the p^+ -i junction interface suppresses the formation of more complex implantation-induced defects during thermal annealing. Together with an enhanced self-diffusion at elevated temperatures[32], the charging effect leads to an increased number of single vacancies available for the formation of NV centers.*”
- XX. Line 221: to address the concerns of the reviewers, the following text has been added: “*Similar effects can also be responsible for the yield enhancement observed after the additional low-energy post-implantation electron irradiation before thermal annealing[35]. The reported enhancement factor of approximately 1.8 is similar to the values obtained in our experiments using the p^+ -i junction structure. Charging of implantation defects may thus be a universal technique to tailor the evolution of implantation damages in the lattice of spin host materials. (new paragraph) The results presented in figure 2 also suggests a preferential path for the formation of NV centers: ...*”.
- XXI. Capture of figure 2: “ $^{15}\text{N}^+$ ” has been corrected to “ ^{15}N ”.
- XXII. Line 281: “*and 10^{10} cm^{-2} ion fluence*” has been added.
- XXIII. The alignment between figures 3 a) and c) is now corrected.

- XXIV. Capture of figure 3: the correct labels have been added (“(circles)” and “(triangles)”); the ion fluence for the CTRIM simulation has been added (“(10^{10} cm^{-2} ion fluence)”).
- XXV. Line 287: “ranging” has been removed.
- XXVI. Line 293: to address the concerns of reviewer 2 and to link the new added section in the Supplementary Information material, the following has been added: “*This remarkable difference in T_2 times is also observed in the statistics of a larger number of NV centers in both areas. Moreover, the enhancement in T_2 times is reproducibly observed for NV centers created using different configurations of the p^+ -i junction structure, e.g. diamonds fabricated with different thicknesses of the boron-doped layer and different nitrogen implantation energies (see SI for further information).*”.
- XXVII. Capture of figure 4: “from residual paramagnetic spins” has been removed.
- XXVIII. Capture of figure 4: to address the concerns of reviewer 2, “*The black dots are the simulated coupling strengths produced by various number of di-vacancy electron spins ($I = 1$) surrounding a central NV center for different distance-radius r_0 of the modeled configuration. Dashed-curves are guides to the eye.*” has been modified to “*The black dots connected by the dashed curves represent the margins of the noise coupling strength calculated for different numbers of di-vacancy electrons spins. These margins were calculated assuming the simplified modeled NV- V_2 configuration depicted in e), which represents residual defects after nitrogen implantation and thermal annealing (see text and SI for details).*”
- XXIX. Line 396: to address the concerns of reviewer 2, “*These values are in good agreement with numerical calculations (see SI) modeling the interaction of a few V_2 spins with an NV center within individual defect-clusters. The calculated coupling strengths are given by black dots in figure 4c related to different numbers of V_2 spins (4 – 2, gray-dotted vertical lines) distributed on a model sphere of radius r_0 around a central NV center, as sketched in figure 4e. The corresponding number of V_2 spins extracted from the experimental results shows that approximately 10 – 30% of all single vacancies within individual defect-clusters build V_2 complexes upon thermal annealing in the reference region, in agreement with the previously outlined MD/KMC simulation values in figure 3c*” has been modified to “*The same figure also shows margins for the coupling strength calculated for a modeled structure comprising a single NV center surrounded by di-vacancy electron spins ($I = 1$). For simplicity, the di-vacancies are assumed to be equally distributed on the surface of a sphere of radius r_0 centered on the NV center, as sketched in figure 4e. This allows us to reduce the variables involved in the calculations to r_0 and the number of di-vacancies only. Values for r_0 and the corresponding number of di-vacancies at different depths correlate to the results from the MD, KMC and CTRIM simulations (see figure 3c and SI). Within the given margins, the calculated results are reasonably confined to the range obtained in our experiments for the additional magnetic noise component in the reference area (figure 4c). This allows us to attribute such noise to the residual implantation-induced defects that are not fully eliminated by the thermal annealing in the reference area.*”.
- XXX. Line 412: the section “Conclusion” now reads “Discussion”.
- XXXI. Line 433: to address the concerns of reviewer 1 regarding the yield enhancement, the following has been added: “(new paragraph) *In the future, the concept of defect charging may be implemented into various methods beyond the suggested p^+ -i junction structure and also combined with different post-implantation techniques such as ion co-implantation[49, 50], possibly increasing the yield of NV centers near the diamond surface even further.*”.
- XXXII. Line 437: “For instance, co-implantation” now reads “Implantation”
- XXXIII. Line 468: “results of” has been added.
- XXXIV. Line 491: “in the p^+ layer” has been added.
- XXXV. Line 492: “of the substrate” has been added.
- XXXVI. Line 520: “presents” now reads “enables”.
- XXXVII. Lines 544 and 546: “therm” has been corrected to “term”.
- XXXVIII. Reference [4]: the eighth author has been corrected to “A. S. Sørensen”.

- XXXIX. Reference [7]: the fifth author has been corrected to “C. T. Schulte-Herbrüggen” and the journal volume and page have been corrected to “506” and “204”, respectively.
- XL. Reference [12]: the sixth author has been corrected to “L. C. L. Hollenberg”.
- XLI. Reference [44] and [53]: the article number has been added.
- XLII. Added references for the co-implantation discussion, as suggested by reviewer 2 (see XXIV): B. Naydenov *et al.*, *Applied Physics Letters* **96**, 163108 (2010); and F. Favaro de Oliveira *et al.*, *Physica Status Solidi (a)* **213**, 2044 (2016).
- XLIII. Added references in the introduction for clarity: M. Widmann *et al.*, *Nat. Mat.* **14**, 164-168 (2015) and T. Kornher *et al.*, *Appl. Phys. Lett.* **108**, 5 (2016).
- XLIV. Acknowledgements: the founding DFG grant has been corrected to “SFB”.

Changes applied to the Supplementary Information:

- I. The number of di-vacancies now reads “ N_{V_2} ”.
- II. Line 6, page 2: “*two body*” has been changed to “*two-body*”.
- III. Line 7, page 4: “*therms*” has been corrected to “*terms*”.
- IV. Line 23, page 5: “*a N di-vacancies*” has been changed to “ *N_{V_2} di-vacancies*”.
- V. Line 3, page 7: “*is show*” has been corrected to “*is shown*”.
- VI. Page 14, capture of figure SI-4: “*an NV centers*” has been corrected to “*an NV center*”.
- VII. Page 15, capture of figure SI-5: “*are show*” has been corrected to “*are shown*”.
- VIII. Page 16, capture of figure SI-6: “ P_i ” has been corrected to “ PI ”.
- IX. To address the concerns of reviewer 2, a new section “*Statistics on spin coherence times*” has been added (see the new document for the complete text). Additionally, supporting data has been added as “*figure SI-7*”.
- X. Added references for the new section (see above): P.-N. Volpe *et al.*, *Diamond and Related Materials* **18**, 1205 (2009), N. Bar-Gill *et al.*, *Nat. Commun.* **3**, 858 (2012) and B. A. Myers *et al.*, *Phys. Rev. Lett.* **113**, 027602 (2014).
- XI. To avoid any misinterpretation regarding the definition of a sample (diamond substrate) and the “sample” area (p^+ -i junction structure after final etching), “sample” now reads everywhere as “diamond”.

REVIEWERS' COMMENTS:

Reviewer #1 (Remarks to the Author):

In my opinion the authors have addressed the comments and suggestions of the reviewers adequately and I suggest publication in Nature Communications in the current form.

Reviewer #2 (Remarks to the Author):

I recommended publication in my first review, subject to some moderate improvements. The careful and detailed response meets those requirements so am happy for the manuscript to be published

OPEN ACCESS Article Processing Charge (APC) Payment Form

JOURNAL: NATURE COMMUNICATIONS

Nature Communications is an open access journal, where all journal content is permanently available free of charge to all readers. Articles are therefore subject to an article processing charge in order to support publication costs (as detailed in the 'Instructions to Authors').

It is mandatory to send this completed form and the Open Access Licence to Publish form to the return address (details below). Your manuscript may not be processed for publication until we have received the relevant forms.

Credit terms are 30 days from receipt of invoice. Failure to pay your invoice within the stated credit term may make you liable to such penalties as restrictions on your ability to publish with NPG or this title in the future, involvement of a third party debt collection agency and legal proceedings.

The APC will be charged at the following rate: £3,150 / \$5,200 / €3,700

VAT or local taxes will be added where applicable: for details please see www.nature.com/TaxInformation

This journal does not have any additional publication fees such as page or colour charges; all authors' costs are included in the article processing charge (APC).

Manuscript Details

Manuscript Number:	<input type="text" value="NCOMMS-16-25843-A"/>
Proposed Title of Article:	<input type="text" value="Tailoring spin defects in diamond by lattice charging"/>
Corresponding Author Name:	<input type="text" value="Felipe Fávaro de Oliveira"/>
Corresponding Author Email Address:	<input type="text" value="f.favaro@physik.uni-stuttgart.de"/>

Billing Details

Contact Name:	<input type="text" value="Jörg Wrachtrup"/>	Organisation Name:	<input type="text" value="3rd Institute of Physics, University of Stuttgart"/>
Email Address:	<input type="text" value="wrachtrup@physik.uni-stuttgart.de"/>	Tel Number:	<input type="text" value="+49 711 685 69832"/>
Billing Address:	<input type="text" value="3rd Institute of Physics
University of Stuttgart
Pfaffenwaldring 57
70569 Stuttgart
Germany"/>		
Postcode/ZIP	<input type="text" value="70569"/>	VAT/GST/Tax Number:	<input type="text" value="DE147794196"/>

Payment Details [Please select one option]

Credit Card [ ]	We will contact you via telephone to obtain your credit card details.
Tel Number (if different from above):	<input type="text" value="Click here to enter text."/>
Invoice [x]	An invoice will be sent by post and email. Payment is required within 30 days.
PO Number (if required)	<input type="text" value="Click here to enter text."/>

PLEASE RETURN YOUR COMPLETED PAYMENT FORM TO:

Email: naturecommunications@nature.com

Macmillan Publishers Limited (trading as Nature Publishing Group).

Registered office: The Campus, 4 Crinan Street, London, N1 9XW, UK. Company number 785998.